# Gene Expressions and High Lymphocyte Count May Predict Durable Clinical Benefits in Patients with Advanced Non-Small-Cell Lung Cancer Treated with Immune Checkpoint Inhibitors

**DOI:** 10.3390/cancers15184480

**Published:** 2023-09-08

**Authors:** Mette T. Mouritzen, Morten Ladekarl, Henrik Hager, Trine B. Mattesen, Julie B. Lippert, Malene S. Frank, Anne K. Nøhr, Ida B. Egendal, Andreas Carus

**Affiliations:** 1Department of Oncology, Aalborg University Hospital, Hobrovej 18-22, 9000 Aalborg, Denmark; morten.ladekarl@rn.dk (M.L.); andreas.carus@rn.dk (A.C.); 2Clinical Cancer Research Centre, Aalborg University Hospital, Sdr. Skovvej 15, 9000 Aalborg, Denmark; anne.noehr@rn.dk (A.K.N.); i.egendal@rn.dk (I.B.E.); 3Department of Clinical Medicine, Aalborg University, Selma Lagerløfs Vej 249, 9260 Gistrup, Denmark; 4Department of Clinical Pathology, Vejle Hospital, University Hospital of Southern Denmark, Beriderbakken 4, 7100 Vejle, Denmarktrine.block.mattesen@rsyd.dk (T.B.M.);; 5Department of Clinical Research, University of Southern Denmark, J.B. Winsløws Vej 19.3, 5000 Odense, Denmark; 6Department of Clinical Oncology and Palliative Care, Zealand University Hospital, Sygehusvej 10, 4000 Roskilde, Denmark; malf@regionsjaelland.dk; 7Department of Clinical Medicine, University of Copenhagen, Blegdamsvej 3B, 2200 Copenhagen, Denmark; 8Center for Clinical Data Science (CLINDA), Aalborg University and Aalborg University Hospital, Sdr. Skovvej 15, 9000 Aalborg, Denmark

**Keywords:** immune checkpoint inhibitors, non-small cell lung cancer, biomarkers, gene expression analysis, lymphocyte count, liver metastases

## Abstract

**Simple Summary:**

Despite the promising results not all patients with advanced non-small cell lung cancer benefit from immunotherapy. Immunotherapy may cause severe adverse events, and therefore, it is important to improve the selection of patients for this treatment. The current biomarker, PD-L1, is a poor predictive biomarker and complementary biomarkers are warranted. This study aimed to assess the association of gene expressions, blood immune cell counts and clinical characteristics with durable clinical benefit of immunotherapy. The findings of this study may assist in the daily clinical assessment of patients with advanced non-small cell lung cancer, that may be candidates for immunotherapy. Additionally, this study could guide future immunotherapy biomarker studies according to methodology and biomarkers of interest, provided a determined effort to collect sufficient biological material. The assessment of PD-L1 by gene expression profiling and the clinical impact of this method compared to standard immunohistochemistry could also be further explored.

**Abstract:**

Background: Not all patients with advanced non-small cell lung cancer (NSCLC) benefit from immune checkpoint inhibitors (ICIs). Therefore, we aimed to assess the predictive potential of gene expression profiling (GEP), peripheral immune cell counts, and clinical characteristics. Methods: The primary endpoint of this prospective, observational study was a durable clinical benefit (DCB) defined as progression-free survival >6 months. In a subgroup with histological biopsies of sufficient quality (*n* = 25), GEP was performed using the nCounter^®^ PanCancer IO 360 panel. Results: DCB was observed in 49% of 123 included patients. High absolute lymphocyte count (ALC) and absence of liver metastases were associated with DCB (OR = 1.95, *p* = 0.038 and OR = 0.36, *p* = 0.046, respectively). GEP showed clustering of differentially expressed genes according to DCB, and a strong association between PD-L1 assessed by GEP (CD274) and immunohistochemistry (IHC) was observed (*p* = 0.00013). The TGF-β, dendritic cell, and myeloid signature scores were higher for patients without DCB, whereas the JAK/STAT loss signature scores were higher for patients with DCB (unadjusted *p*-values < 0.05). Conclusions: ALC above 1.01 × 10^9^/L and absence of liver metastases were significantly associated with DCB in ICI-treated patients with NSCLC. GEP was only feasible in 20% of the patients. GEP-derived signatures may be associated with clinical outcomes, and PD-L1 could be assessed by GEP rather than IHC.

## 1. Introduction

Immune checkpoint inhibitors (ICIs), anti-Programmed Death-(Ligand)-1 (PD-(L)1) antibodies have revolutionised the treatment of patients with advanced non-small cell lung cancer (NSCLC). Randomised controlled trials (RCTs) have demonstrated improved overall response rates, progression-free survival (PFS), and overall survival (OS) compared to standard chemotherapy in patients treated with first- or subsequent-line ICI monotherapy [1,2,3,4,5]. Furthermore, a subgroup of patients becomes long-term responders with improved 3- and 5-year survival rates in both RCTs and daily cancer care [6,7,8]. In some countries, including Denmark, the selection of patients with advanced NSCLC for ICI-based treatment depends on the PD-L1 tumor proportion score (TPS) [9]. However, PD-L1 TPS has shown limited potential as a single predictive biomarker of response to ICIs. In patients with squamous NSCLC treated with ICI in subsequent lines (≥2 L), no significant survival differences between PD-L1 negative and PD-L1 positive patients were observed, and around 40% of patients treated with ICI in the first line (1 L) having PD-L1 TPS ≥90% do not respond [1,10]. Therefore, complementary biomarkers have been proposed and may be related to both tumor cells, the tumor microenvironment (TME), the immune system, and other host factors. Besides PD-L1 TPS, microsatellite instability (MSI)/mismatch repair deficiency and tumor mutational burden (TMB) have been approved as ICI biomarkers by the Food and Drug Administration (FDA) [11,12]. Due to the continuous and dynamic nature of TMB, no gold-standard method or cut-off value exists [13]. Consensus guidelines exist for MSI due to its clinical role in cancers associated with Lynch syndrome [14]. However, in NSCLC, the prevalence of MSI-high and TMB-high status is only approximately 1% and 15%, respectively, and they are not yet incorporated into clinical guidelines in NSCLC [15,16]. 

Other comprehensively investigated clinical factors with prognostic value and a possible association with ICI efficacy include the immune phenotypes, the presence of tumor-infiltrating lymphocytes (TILs), and their relative abundance and location [17]. In addition, an INFγ-related 18-gene mRNA, T-cell inflamed gene expression signature (TIS), has been associated with improved ICI response across different tumor types [18]. The 18-gene TIS was also applied to The Cancer Genome Atlas (TCGA) RNA-sequencing dataset, showing high median TIS scores in NSCLC resections [19]. Gene expression profiling (GEP) holds the potential to integrate the investigation of biomarkers related to tumor cells, TME, and immune cells simultaneously. Furthermore, GEP can be performed with a relatively low amount of RNA with good quality and hence should not require large tissue samples or resections [18]. Exploratory GEP has been performed in some previous RCTs, and potential predictive gene expression signatures have been proposed across different tumor types and ICI treatment regimens [5,20,21,22,23]. However, few studies of GEP in routine clinical practice of patients with advanced NSCLC have been conducted [24]. Peripheral blood biomarkers have been associated with ICI efficacy, such as neutrophil-to-lymphocyte-ratio (NLR), lactate dehydrogenase, and absolute lymphocyte count (ALC) [25,26]. A post hoc analysis of the phase III OAK trial showed predictive value of NLR in ICI-treated patients compared to patients treated with chemotherapy [27]. Furthermore, high pre- and post-ICI treatment ALC has been associated with improved survival in patients with NSCLC [28]. To increase the predictive value, and hence to improve the selection of patients for ICI treatment, different immunograms and models have included multiple of the proposed biomarkers [29,30,31]. Although none of the proposed biomarkers or predictive models have been implemented in the clinical treatment guidelines for patients with NSCLC, the clinical variables and gene expression signatures have shown promising predictive potential.

In this study, we aimed to assess the impact of gene expressions, clinical features, and peripheral immune cell counts on durable clinical benefit (DCB) in patients with advanced NSCLC treated with ICIs in routine clinical cancer care.

## 2. Materials and Methods

### 2.1. Study Design and Patients

The study was a prospective, observational, and explorative study. The study population consisted of consecutively included patients with advanced NSCLC who received at least one cycle of anti-PD-1 or anti-PD-L1 monotherapy as 1 L or ≥2 L of treatment. Patients with EGFR mutations, ALK rearrangements, or curative treatment options were excluded.

At the Department of Oncology, Aalborg University Hospital, 58 patients were included, regardless of treatment line, from August 2018 to September 2019 (ClinicalTrials.gov NCT03658460). An additional cohort of 65 consecutive patients treated with 1 L ICI was included at the Department of Clinical Oncology and Palliative Care, Zealand University Hospital, Naestved, from July 2018 to June 2020 (ClinicalTrials.gov NCT03512847). Treatment criteria, monitoring, and follow-up were similar at the two recruiting departments.

### 2.2. Data Collection and Data Management

Baseline characteristics were prospectively collected, including age, sex, eastern cooperative oncology group (ECOG) performance status (PS), smoking status, BMI, and TNM stage (IASLC 8th edition) with additional information on metastatic sites in case of stage IV disease. Furthermore, information on biopsy modalities, NSCLC histopathological subtype, and PD-L1 TPS was recorded. From baseline peripheral blood samples, the ALC and absolute neutrophil count (ANC) were obtained, and the NLR was derived (ANC/ALC). 

Patients received ICI treatment according to the national treatment guidelines at that time: pembrolizumab 2 mg/kg/3 w or 200 mg/3 w, atezolizumab 1680 mg/4 w, or nivolumab 3 mg/kg/2 w [32,33]. Information describing the patient’s treatment was collected, which included treatment line, treatment duration, reasons for ICI discontinuation, ICI treatment beyond progression, and post-ICI systemic antineoplastic treatment. 

CT scans were performed every 8–9 weeks for treatment response evaluation according to the Response Evaluation Criteria in Solid Tumors (RECIST) version 1.1. The primary clinical endpoint was durable clinical benefit (DCB), defined as progression-free survival (PFS) > 6 months. PFS was calculated from the first ICI administration date (index date) to the date of progressive disease (PD), death, or the last follow-up or censoring date. The last follow-up date was defined as the date of the last radiological response evaluation. No patients were lost to follow-up. Furthermore, OS was calculated from the index date to the date of death or the date of data cut-off. The censoring date was 1 March 2022 for patients treated at Zealand University Hospital and 1 May 2022 for patients treated at Aalborg University Hospital. 

### 2.3. Tissue Samples and Routine Diagnostics

The tissue samples were routinely processed histological or cytological diagnostic biopsies, formalin-fixed and paraffin-embedded (FFPE). In most of the patients receiving ≥2 L ICI, systemic antineoplastic treatment was administered between the time of tissue sampling and the date of first ICI administration (*n* = 25; 93%). The routine diagnostic framework included morphological examination and immunohistochemistry (IHC) to establish the cancer diagnosis and determine the histopathological subtype of NSCLC. Standard assessment of PD-L1 TPS was performed by IHC with the 22C3 pharmDx antibody stained on the Dako Omnis platform. PD-L1 TPS was categorised as <1%, 1–49%, and ≥50% of tumor cells with staining for membranous PD-L1 [34]. Next-generation sequencing (NGS) was routinely performed with the TruSight^®^ Tumor 15 assay (Illumina, San Diego, USA) (patients included at Aalborg University Hospital) or GeneRead QIAact AIT Panel (patients included at Zealand University Hospital, Naestved, Denmark) to assess EGFR, BRAF, KRAS and ERBB2 status. ALK rearrangements were routinely assessed by IHC, and in cases with inconclusive or positive IHC, additional fluorescence in situ hybridisation (FISH) was performed to confirm the presence/absence of ALK rearrangements.

### 2.4. Gene Expression Profiling

Prior to GEP, the tumor percentage was estimated in histological samples by a pathologist. After excluding patients with cytology only, insufficient tissue, failing quality controls (QC), or failed analysis, the final GEP-cohort consisted of 25 patients (Figure 1).

Histological samples were analysed using a 770-gene expression panel, the nCounter^®^ PanCancer IO 360 panel (NanoString Technologies, Inc., Seattle, WA, USA). According to the recommendations of the manufacturer, extraction of total ribonucleic acid (RNA) was performed manually on 10 × 5 µm sections from FFPE samples using the miRNeasy^®^ FFPE kit (Qiagen, Venlo, The Netherlands). The extracted RNA was eluted in 13 µL RNAase-free water, and the RNA concentrations were determined by using the Qubit 3 Fluorometer (Invitrogen^TM^, Thermo Fischer Scientific, Carlsbad, CA, USA). The purified RNA was stored at −80 °C. Only samples with an RNA concentration ≥ 60 ng/ul were included in the final GEP cohort. An input amount of 300 ng RNA was used for each sample during NanoString analysis. Hybridisation was performed using the nCounter^®^ PanCancer IO360 gene expression panel (NanoString Technologies, Inc.). The technical integrity of the nCounter^®^ profiling assay underwent further QC assessment. The sample input and reaction efficiency were assessed by the geometric mean of housekeeper genes in each sample. A minimum geometric mean count of 32 housekeeper genes was required for analysis, and geometric mean counts of 32–100 were considered borderline. Furthermore, the nCounter^®^ profiling assay was assessed according to imaging, binding density, positive control linearity, and limit of detection. To correct for cartridge differences, background correction and data normalisation were performed before the final data analysis. The final analysis included data from samples that passed all QC steps.

### 2.5. Next Generation Sequencing

TMB and MSI status was assessed by NGS. DNA was extracted from 10 × 5 µm sections from FFPE samples using the Maxwell^®^ 16 FFPE Plus LEV DNA Purification Kit (AS1135). Only samples with a DNA concentration ≥3.33 ng/ul were included in the final GEP cohort. The TruSight^®^ Oncology 500 (TSO500; Illumina, SD, USA) gene panel was used for sequencing analysis. Library preparation was performed using the TruSight^®^ Oncology 500 reagent kit according to the manufacturer’s protocol, and the samples were run on the NextSeq^TM^ 550 instrument (Illumina^®^) [35]. Only samples that passed all sequencing QCs were included for further analysis. The TSO500 Local Run Manager TruSight^®^ Oncology 500 v2.2 Analysis Module was used to generate TMB and MSI scores [36]. TMB was defined as the number of eligible variants divided by the effective panel size. The TMB-high cut-off was 10 mutations/Mb. The MSI score was defined as the number of unstable MSI sites divided by the total number of assessed MSI sites [36]. The MSI-high cut-off was 20%. 

### 2.6. Statistical Analyses

#### 2.6.1. Descriptive Statistics, Logistic Regression, and Survival Analyses

Comparisons of patients receiving ICI treatment in 1 L or ≥2 L were performed with ANOVA tests for the continuous variables and Fisher’s exact tests for the categorical variables. Fisher’s exact test was chosen to account for the low expected values. Median values of ALC, ANC, and NLR were used for the comparisons.

Logistic regression analysis was used to assess factors associated with DCB. First, univariable logistic regression analyses were conducted with DCB as the dependent variable and each of the baseline characteristics as the independent variable. Brain-, bone-, and liver metastases were included as the only metastatic sites due to the known prognostic impact on survival in NSCLC. Secondly, multivariable logistic regression analysis was conducted and included age, sex, PS, PD-L1, and factors significantly associated with DCB in the univariable logistic regression analysis. Wald test *p*-values and profile likelihood confidence limits were reported.

A Cox proportional hazards model was used for the OS analysis. Analyses were restricted to patients receiving 1 L ICI treatment (*n* = 96) due to significant differences in selection criteria for ICI and prognostic clinical and pathological factors according to treatment line. Univariable Cox regression analyses were performed for baseline characteristics. Subsequently, a multivariable Cox regression analysis was performed, including age, sex, PS, and factors significantly associated with OS in the univariable analyses. Schoenfeld residuals revealed no significant nonproportionality in the multivariable model, indicating that the assumption of proportional hazards was reasonable. 

A ROC curve using ALC as a predictor for DCB was drawn. A two-sample Kolmogorov–Smirnov plot was used to find the optimal ALC cut-off for predicting DCB [37]. This optimal cut-off was determined as the cut-off value of the ALC that yielded the maximal difference between the cumulative density of ALC in the DCB negative/DCB positive group. 

*p*-values < 0.05 were considered statistically significant, and no adjustments for multiple testing were performed. Statistical analyses were performed with R version 4.2.1 [38].

#### 2.6.2. Bioinformatics

##### Differential Expression of Genes

Gene expression analyses were performed to identify differentially expressed genes for response (DCB vs. no DCB). First, gene counts were normalised to log2 counts per million using the function Voom (Limma R package) and the trimmed mean of M-values (TMM) method from the R package edgeR [39,40]. Next, a linear model was fit to each gene, adjusting for biological factors associated with DCB using the R package limma [39]. The *p*-values were corrected for multiple testing using Benjamini–Hochberg false discovery rate (FDR). Significant differentially expressed genes were identified using FDR cutoff of 5%. The patterns of the gene expression of genes with a *p*-value below 0.05 were further explored using the ComplexHeatmap package [41]. The package was applied to cluster the patients and the genes using hierarchical clustering based on Euclidean distance. ANOVA test was used to assess the association between the categorical IHC-derived PD-L1 TPS and the continuous GEP-derived PD-L1 (CD274).

##### Gene Expression Signatures 

Differences in gene expression signature scores according to DCB were also evaluated. Gene expression signature scores were calculated as a weighted linear combination of the included genes’ expression values normalised to stable housekeeper gene expression as described by the manufacturer [18]. As in the gene expression analysis, a linear model was fit to each gene, adjusting for biological factors associated with DCB using the R package limma [39]. The *p*-values were using FDR, and FDR < 0.05 was considered statistically significant.

## 3. Results

### 3.1. Baseline Patient Characteristics 

The study included 123 patients; 25 patients (20%) with GEP and 98 patients (80%) without GEP. Overall, 44% of patients were female, and the median age was 67 years (range: 46–86). The comparison of baseline characteristics and peripheral immune cell counts in patients with and without GEP showed significantly more squamous cell carcinomas in the GEP cohort (*p* = 0.007) (Table 1). 

### 3.2. Treatment Characteristics 

ICI was administered in 78% of the patients as 1 L and in 22% as ≥2 L. Significant differences in PS, PD-L1 TPS, NSCLC histopathological subtype, lung metastases, and peripheral lymph node metastases were observed between patients treated with 1 L and ≥2 L ICI. No significant differences in median ALC (*p* = 0.33), ANC (*p* = 0.84), and NLR (*p* = 0.21) were observed according to the treatment line (Appendix A). The median time to treatment discontinuation was 105 days (range 1–763), without significant differences between 1 L and ≥2 L (*p* = 0.14). ICI treatment was discontinued due to PD (*n* = 68; 55%), toxicity (*n* = 33; 27%), completion of 2 years of ICI treatment (*n* = 10; 8%), poor PS (*n* = 8; 7%), death (*n* = 4; 3%), and/or ‘other’ reasons (*n* = 19; 15%). ‘Other reasons’ included lack of compliance, patient’s choice, comorbidity, or high dose steroid. ICI treatment could be discontinued due to more than one reason. Systemic antineoplastic treatment after ICI discontinuation was administered in 49% of the patients (*n* = 60), without statistically significant difference according to treatment line. Treatment beyond PD was observed in 11 patients (9%). A swimmer plot showing the course of individual patients from the initiation of ICI treatment is shown in Appendix A. 

### 3.3. Predictive Factors of Durable Clinical Benefit 

DCB was observed in 49% (*n* = 60) of all patients and did not significantly differ in 1 L compared to ≥2 L (51% vs. 41%, *p* = 0.40). A comparison of patients with and without DCB showed that the presence of liver metastases was significantly associated with not achieving DCB (30% vs. 12%, *p* = 0.02), and ALC above the median was significantly more frequent in patients with DCB (*p* = 0.01). 

Likewise, in the univariable logistic regression analysis, liver metastases (OR 0.31, *p* = 0.01) and ALC (OR 2.05, *p* = 0.02) were significantly associated with DCB (Figure 2 and Appendix A). In multivariable logistic regression analysis, liver metastases (*p* = 0.046) and ALC (*p* = 0.038) remained significantly associated with DCB (Figure 2). The increased rate of DCB in patients with PD-L1 ≥1% did not reach statistical significance (Figure 2).

A ROC curve analysis was made to investigate the predictive potential of ALC as a single biomarker for DCB, and this yielded an AUC of 0.63 (Appendix A). An optimal cut-point of 1.01 × 10^9^/L was found, corresponding to the 25% quartile, and using ALC dichotomised at this cut-point as a predictive biomarker for DCB resulted in a false positive rate of 0.64 and true positive rate of 0.90. DCB was observed in 21% of all patients with an ALC below the optimal cut-point of 1.01 × 10^9^/L and in 57% of all patients with an ALC above the optimal cut-point (Figure 3).

The mOS was 19.2 months (95%CI 0.33–41.7) and 12.5 months (95%CI 0.16–40.8) in patients treated with 1 L and ≥2 L ICI, respectively (*p* = 0.09). Increased ALC was also associated with improved OS in multivariable Cox regression analysis of patients treated with 1 L ICI (Appendix A).

### 3.4. The GEP Subpopulation

#### 3.4.1. Treatment Characteristics and Clinical Outcomes

GEP was feasible in 33% (*n* = 25) of all patients with diagnostic histological biopsies (*n* = 74) (Figure 1). Significantly more patients with GEP received ICI in ≥2 L (*p* = 0.03) compared to patients without GEP. No significant differences in time to treatment discontinuation, DCB, and mOS were observed between the patients with and without GEP (Appendix A). 

#### 3.4.2. Gene Expression Analyses 

A comparison of gene expressions between patients with DCB and without DCB revealed 53 genes with a *p*-value below 0.05 (Appendix A). PD-L1 (CD274) was one of those genes (*p* = 0.03); however, no genes were significant after adjustment for multiple testing (no FDR below 0.05). Pearson correlation of PD-L1 with genes differentially expressed between DCB and no DCB showed a significant negative correlation with LTBP1 (*p* < 0.05) and a positive correlation with TAP1 and ITGAE (*p* < 0.05). A highly significant association between the categorical PD-L1 TPS assessed by IHC and the continuous GEP-derived PD-L1 (CD274) was identified (*p* = 0.00013). Furthermore, PD-L1 (CD274) was differentially expressed between patients receiving 1 L and ≥2 L ICI (*p* = 0.0017), reflecting the treatment inclusion criteria (Figure 4).

The patterns of the expression of the 53 genes with a *p*-value < 0.05 were explored, and hierarchical clustering showed that two clusters separated the patients with and without DCB except for two patients. An intermediary heterogeneous cluster consisted of patients with or without DCB (Figure 5). 

The gene expression signature scores in patients with DCB and without DCB were compared. These analyses identified no signatures with FDR < 0.05; however, four signatures had an unadjusted *p*-value < 0.05. The TGF-β (*p* = 0.047, log2FC = −0.92), dendritic cell (DC) (*p* = 0.025, log2FC = −0.92), and myeloid (*p* = 0.024, log2FC = −0.80) signature scores were higher for patients without DCB whereas the JAK/STAT loss signature scores (*p* = 0.005, log2FC = 1.41) were higher for patients with DCB. 

### 3.5. Next Generation Sequencing 

TMB and MSI were available in only 42% (*n* = 51) of all the patients (*n* = 123), and 47% (*n* = 24) of the analysed tissue samples were TMB-high. NGS was feasible in only 24% (*n* = 6) of patients in the GEP subpopulation. No tumor samples were MSI-high, and therefore, MSI status was not included in the analyses. 

## 4. Discussion

This prospective study included 123 consecutive patients with advanced NSCLC treated with ICI in routine clinical cancer care. The association of baseline characteristics, peripheral immune cell counts, and GEP was assessed, with DCB being the primary clinical endpoint. No consensus on the DCB definition exists, and we defined DCB as PFS > 6 months to increase comparability with other GEP studies in NSCLC [42,43]. The DCB was similar regardless of treatment line, allowing for analysis of predictive factors for DCB in the combined population. Additionally, the time to treatment discontinuation, as a proxy for dose intensity, and mOS were similar regardless of treatment line. 

As demonstrated in other cohort studies and RCTs, the presence of liver metastases was negatively associated with DCB and OS [44,45,46]. A study of patients with malignant melanoma showed that liver metastases had significantly lower T-cell infiltration and increased TIM-3 expression than lung and lymph node metastases [47]. A recent study in NSCLC also demonstrated that the CD8+ T-cell infiltration was lower in liver metastases compared to other metastatic lesions and that combined PD-L1 TPS ≥ 1% and CD8+ T-cell infiltration in liver metastases increased PFS [48]. These biological mechanisms may contribute to the poorer ICI efficacy in patients with liver metastases. 

An increase in ALC was significantly associated with DCB and improved OS in our study. We also found that an ALC of 1.01 × 10^9^/L was the most optimal cut-point for predicting DCB, and confirmatory studies in independent, larger populations are warranted. High pre- and post-ICI peripheral lymphocyte counts and specific subsets of peripheral lymphocytes have also been associated with improved outcomes in ICI-treated patients with NSCLC, whereas lymphopenia has been associated with impaired survival [28,49,50]. Additionally, a lower percentage of peripheral lymphocytes in NSCLC has been observed in male patients and patients with bone- and liver metastases and has been associated with poor survival regardless of NSCLC histopathological subtype and disease stage [51]. 

In contrast, no association between ANC or NLR and DCB or OS was found in our study. A meta-analysis showed that a higher NLR was associated with poorer OS in ICI-treated patients with lung cancer; however, other factors, such as the NSCLC subtype, may impact the predictive value of NLR [52]. A recent study demonstrated that lung adenocarcinomas had more effector and activated T cells and fewer Treg cells compared to lung squamous cell carcinomas assessed by single-cell RNA sequencing from surgical resections [53]. 

NSCLC histopathological subtype and lymphocyte counts may interact and impact the GEP, and therefore, the analyses of gene and gene-expression signatures were adjusted for NSCLC histopathological subtype and ALC. However, no significant differentially expressed genes or signatures were found when adjusting for multiple testing (no FDRs below 0.05). Despite the lack of significant FDRs, a clustering tendency of differentially expressed genes was observed according to DCB (Figure 4) and indicates that certain gene expression phenotypes may be associated with DCB in ICI-treated patients with advanced NSCLC. 

Notably, a strong association between the categorical PD-L1 TPS assessed by IHC and the continuous GEP-derived PD-L1 (CD274) was identified. Furthermore, PD-L1 (CD274) was differentially expressed between patients receiving 1 L and ≥2 L ICI, which correspond to the different PD-L1 cut-offs in treatment guidelines in patients with advanced NSCLC [9]. These findings indicate the clinical relevance of GEP in treatment decisions, and GEP eliminates the intra- and inter-observer differences in the IHC assessment of PD-L1 [54]. 

Most NSCLC gene expression studies rely upon surgically resected early-stage tumors with large tissue resections. They, thus, may not be comparable to gene expressions in ICI-treated patients with advanced-stage disease [19,55]. In the POPLAR and IMpower 150 studies, effector-T cell gene expression signatures were associated with ICI efficacy [5,20]. However, due to major differences in GEP assays, gene expression signature definitions, patient populations, and treatment regimens, a direct comparison with our study results is not reasonable. Only a few GEP studies have included patients with advanced NSCLC treated with ICI in routine clinical care. One other study used the same panel and included both histological and cytological samples [42]. However, no comparison between the two sample types was performed according to RNA quality or differences in intratumor gene expressions [42]. Two GEP studies in ICI-treated patients with NSCLC have shown that DC66b expression was associated with poor OS [56] and high immune cell scores (T cells, NK-cells, and M1 macrophages) were associated with DCB, respectively [42,43]. Another study found INF- and antigen processing/presentation signatures to be positively associated with PFS in adenocarcinomas and TME signatures to be associated with PFS in squamous cell carcinomas [57]. However, adjustments for baseline clinicopathological factors prior to the gene expression analyses have not been consistently performed, which may confound the true predictive association with ICI efficacy of the proposed gene expressions. Furthermore, the comparison of different GEP studies is very challenging due to the small sample sizes and differences in disease stage, NSCLC subtypes, tissue sample types, sample sites (primary or metastatic, and which metastatic location), RNA preparation, GEP panels, statistical analyses, and endpoint definitions. Currently, no large GEP datasets of advanced-stage NSCLC cancer cohorts are available for validation. 

In our study, only 25 patients were included in the GEP cohort. This low number of suitable samples was primarily due to insufficient material (30/74 samples). Low concentration of RNA was the cause of exclusion in around one-fifth of the histological samples (16/74 samples), which may be explained by the thin sections (5 µm) holding a lower percentage of intact cells and more fragmented RNA compared to larger sections (10–20 µm) [58]. Therefore, the implementation of GEP in routine diagnostics primarily requires a revision of the diagnostic framework, as suggested by Hirsch et al. [59]. In the GEP cohort, most patients treated with ≥2 L ICI had received chemotherapy between the time of the diagnostic tissue sampling and ICI initiation. A pre- and post-chemotherapy gene expression analysis of 29 paired samples has shown that the average expression of CTLA4, LAG3, TNFRSF18, CD80, and FOXP3 in an immune module was significantly decreased in post-chemotherapy samples, and dynamic changes in INF-γ expression were observed [60]. Additionally, INF-γ expression has been associated with improved outcome in ICI-treated patients [18,60,61]. However, the impact of previous chemotherapy on ICI efficacy has not been assessed in clinical cohorts.

No gene expression signatures were significantly associated with DCB when adjusting for multiple testing. However, without adjustment, DC, myeloid, and TGF-β signature scores were higher in patients without DCB, and JAK/STAT loss signature scores were higher in patients with DCB. 

DCs are antigen-presenting cells that contribute to the initiation of anti-tumoral T-cell responses [62]. However, immature DCs generally induce immune tolerance, and tumors may induce immune evasion by disruption of normal DC function [62,63]. The GEP performed in our study did not assess the functional status of DCs. Myeloid cells in the TME include tumor-associated macrophages (TAMs), tumor-associated neutrophils (TANs), and myeloid-derived suppressor cells (MDSCs), which promote tumor cell growth and invasion and suppress immune responses [64,65]. TGFβ (TGFB1) in the TME inhibits immune activity against tumors and promotes tumor growth and survival [66]. Clinical investigation of dual inhibition of TGFβ and PD-(L)1 is ongoing in many solid tumors, including NSCLC [67].

The JAK/STAT pathway plays an essential role in the differentiation of T-helper cells, and JAK/STAT inhibition in Tregs has shown downregulation of Foxp3, which attenuates the immunosuppressive function [68,69]. Hence, the JAK/STAT function is cell-specific, and the impact of JAK/STAT loss on ICI efficacy seems to be cell-dependent. However, in our study, the JAK/STAT loss signature, defined by the manufacturer, was not restricted to a specific cell type, and cell type-specific analyses are warranted to investigate the association with ICI efficacy.

### Strengths and Limitations

The main strength of this study was the clinical relevance according to the target population (advanced/metastatic NSCLC) and treatment setting (palliative ICI treatment). Furthermore, patients were consecutively included, clinical data completeness was high, and no patients were lost to follow-up. Only histological biopsies were used for GEP to increase the likelihood of the tumor, TME, and immune response biology to be represented in the samples. Additionally, RNA amplification was not performed to avoid the risk of amplification bias.

The main limitation of this study was the low number of patients included in the GEP cohort (*n* = 25) due to the large proportion of histological samples with insufficient material or low RNA concentration. Furthermore, the impact of chemotherapy on gene expressions prior to ICI treatment was not assessed. The potential interaction between peripheral ALC and other baseline characteristics, such as NSCLC histopathological subtype, metastatic sites, sex, and age, was also not assessed in this study.

## 5. Conclusions

In patients with advanced NSCLC treated with ICIs in routine clinical cancer care, high ALC and absence of liver metastases were significantly associated with DCB. PD-L1 assessed by GEP was highly correlated with IHC-assessed PD-L1 and treatment line, indicating a clinical relevance of GEP for less biased PD-L1 estimation. DC, myeloid, and TGF-β signature scores were higher in patients without DCB, and JAK/STAT loss signature scores were higher in patients with DCB. However, with the current routine diagnostic framework, GEP is only feasible in one-third of the patients.

## Figures and Tables

**Figure 1 cancers-15-04480-f001:**
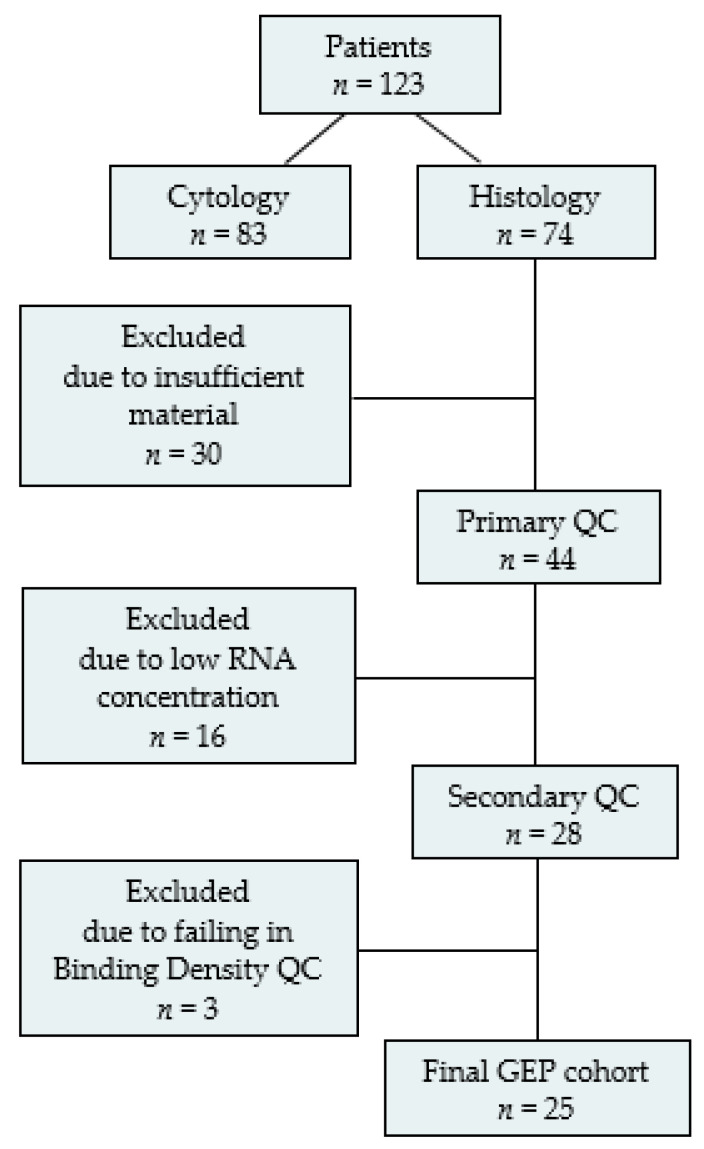
Flowchart of baseline tissue samples prior to gene expression profiling. The trajectory of tissue samples selected for gene expression profiling (GEP). *n*, number of patients; QC, quality control.

**Figure 2 cancers-15-04480-f002:**
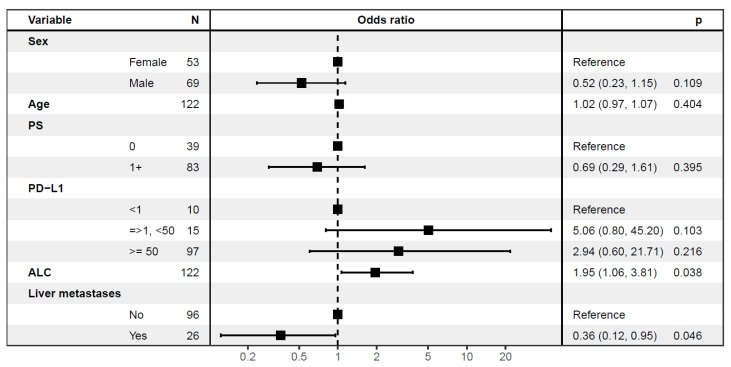
Multivariable logistic regression analysis assessing the association between baseline characteristics and durable clinical benefit. Multivariable logistic regression showing significant positive association between high median absolute lymphocyte count (ALC) and durable clinical benefit (DCB) and negative association between liver metastases and DCB. N, number of patients; PS, performance status; PD-L1, programmed death-ligand 1.

**Figure 3 cancers-15-04480-f003:**
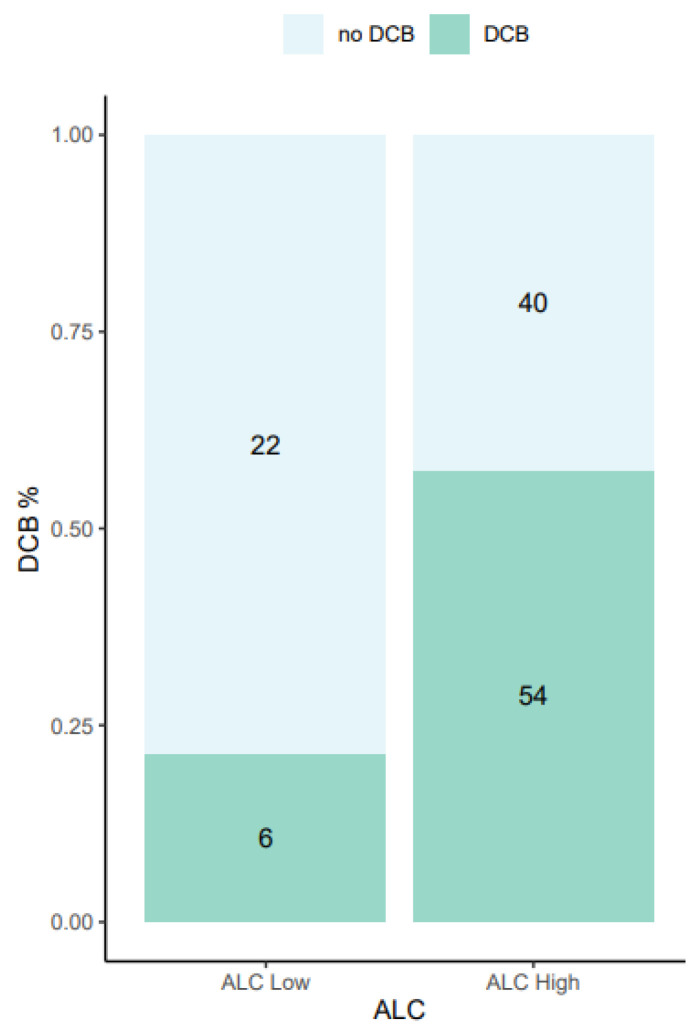
Bar chart presenting the relationship between absolute lymphocyte count and durable clinical benefit. All patients (*n* = 122) were categorised as ALC low or ALC high, separated by the optimal ALC cut-point of 1.01 × 109/L. The numbers in the bars represent the absolute number of patients in each group. ALC was missing in one patient. DCB, durable clinical benefit; ALC, absolute lymphocyte count.

**Figure 4 cancers-15-04480-f004:**
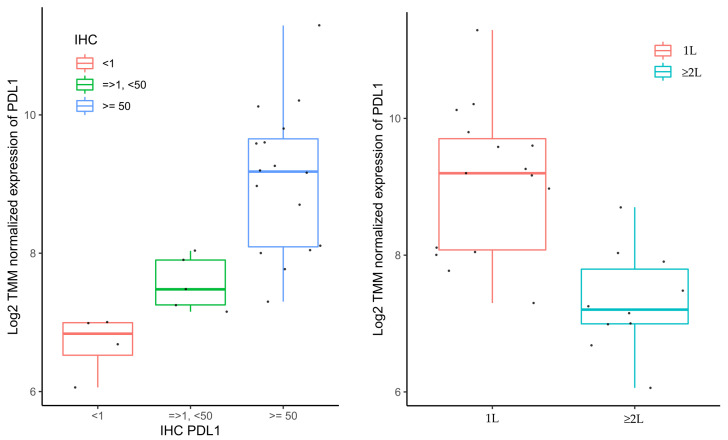
The association between PD-L1 (CD274) derived by gene expression profiling and PD-L1 assessed by immunohistochemistry and treatment line. Boxplots of log2 normalised expression of PD-L1 for three levels of PD-L1 assessed by IHC (*p* = 0.00013) (to the left) and treatment line (*p* = 0.00017) (to the right). PDL1, programmed death-ligand 1; IHC, immunohistochemistry; 1 L, first-line treatment; ≥2 L, treatment in second or subsequent line.

**Figure 5 cancers-15-04480-f005:**
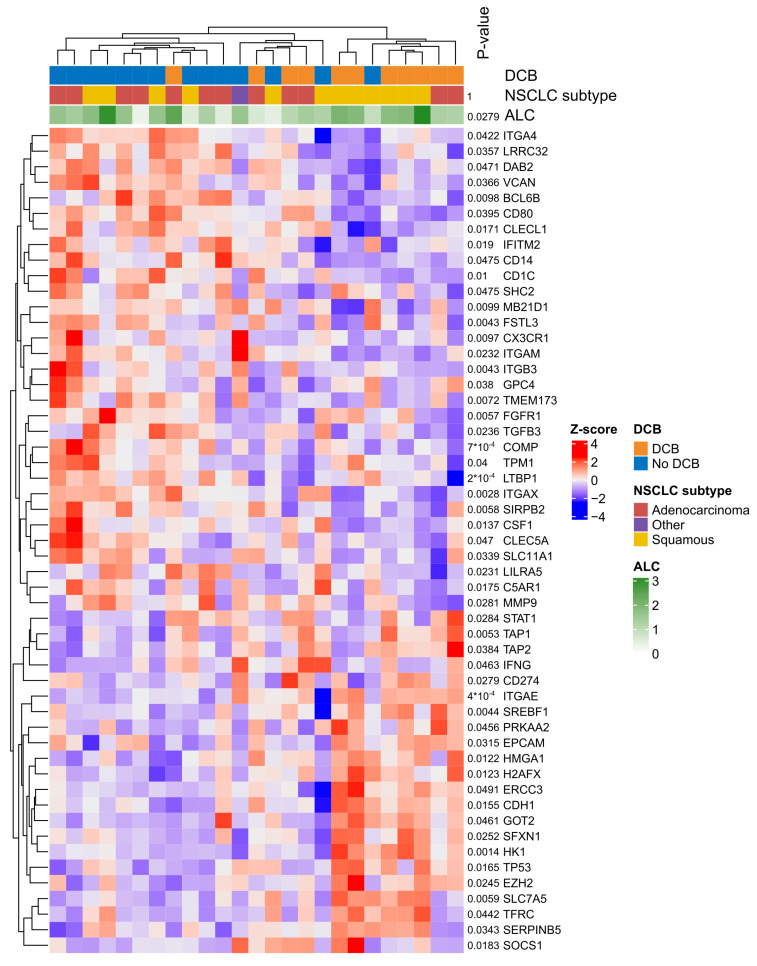
Hierarchical clustering of genes with a *p*-value below 0.05. Heatmap of gene expression z-scores for genes with a *p*-value < 0.05 in comparison between DCB vs. no DCB. The patients (columns) (*n* = 25) and the genes (rows) are clustered using hierarchical clustering based on Euclidean distance. The dendrogram added to the top and to the left visualise the order of the clustering. In the top three, annotation rows are added to indicate each patient’s DCB status, NSCLC subtype, and ALC. Finally, a *p*-value is listed for each row. The *p*-value for NSCLC subtype and ALC compares DCB vs. no DCB using a Fisher’s exact test and unpaired *t*-test, respectively. The *p*-value in front of the genes derives from the gene expression test.

**Table 1 cancers-15-04480-t001:** Baseline characteristics and peripheral blood values of patients with and without gene expression profiling.

Baseline Characteristics	GEP *n* (%)	No GEP *n* (%)	Total *n* (%)	*p*-Value
Patients	25 (20)	98 (80)	123 (100)	
Age, median years (range)	68 (52–82)	67 (46–86)	67 (46–86)	0.90
Sex				
Male	12 (48)	57 (58)	69 (56)	
Female	13 (52)	41 (42)	54 (44)	0.38
Performance status				
0	7 (28)	32 (33)	39 (32)	
1	11 (44)	53 (54)	64 (52)	0.23
≥2	7 (28)	13 (13)	20 (16)	
Smoking status				
Current	9 (36)	33 (34)	42 (34)	
Former	16 (64)	63 (64)	79 (64)	1
Never	0 (0)	2 (2)	2 (2)	
BMI, median (range)	25 (17–39)	24 (16–41)	24 (16–41)	0.78
TNM stage				
III	3 (12)	14 (14)	17 (14)	
IV	22 (88)	84 (86)	106 (86)	1
Metastatic sites ^a^				
Brain	3 (12)	7 (7)	10 (8)	0.42
Bone	8 (32)	25 (26)	33 (27)	0.61
Liver	8 (32)	18 (18)	26 (21)	0.17
Adrenal glands	3 (12)	27 (28)	30 (24)	0.12
Distant lymph nodes	5 (20)	11 (11)	16 (13)	0.32
Lung	8 (32)	25 (26)	33 (27)	0.61
Pleura ^b^	7 (28)	36 (37)	43 (35)	0.49
Soft tissue ^c^	4 (4)	4 (4)	5 (4)	1
Other	2 (8)	22 (22)	24 (20)	0.16
NSCLC subtype				
Adenocarcinoma	12 (48)	72 (74)	84 (68)	
Squamous cell carcinoma	12 (48)	17 (17)	29 (24)	0.007
Other ^d^	1 (4)	9 (9)	10 (8)	
PD-L1				
<1%	4 (16)	6 (6)	10 (8)	
≥1% and <50%	5 (20)	10 (10)	15 (12)	0.08
≥50%	16 (64)	82 (84)	98 (80)	
TMB				
High	4 (16)	20 (20)	24 (20)	
Low	2 (8)	25 (26)	27 (22)	0.40
Missing	19 (76)	53 (54)	72 (58)	
Blood values, median (range) *				
ALC	1.40 (0.30–3.15)	1.36 (0.30–3.60)	1.40 (0.30–3.60)	0.83
ANC	7.30 (3.79–16.2)	6.50 (2.90–36.3)	6.70 (2.90–36.3)	0.70
NLR	5.16 (1.50–37.7)	4.31 (1.16–34.7)	4.40 (1.16–37.7)	0.06

^a^ Patients may be registered with more than one metastatic site. Each metastatic site was recorded as a categorical variable (yes or no), and the *p*-values reflect the distribution of the two levels for each metastatic site. ^b^ ‘Pleura’ included pleural fluid. ^c^ ‘Soft tissue’ included cutis, subcutis, and muscles. ^d^ ‘Other’ included NSCLC not otherwise specified (NOS) and sarcomatoid carcinoma. GEP, gene expression profiling; *n*, number of patients; BMI, body mass index; TNM, tumor-node-metastasis classification of malignant tumors; NSCLC, non-small cell lung cancer; PD-L1, programmed death-ligand 1; TMB, tumor mutational burden; ALC, absolute lymphocyte count; ANC, absolute neutrophil count; NLR, neutrophil-to-lymphocyte ratio. * ALC and NLR were missing in one patient in the ‘No GEP’ cohort.

## Data Availability

Not applicable.

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
