# Peer review of "Gene Expressions and High Lymphocyte Count May Predict Durable Clinical Benefits in Patients with Advanced Non-Small-Cell Lung Cancer Treated with Immune Checkpoint Inhibitors"

_cancers, 2023, doi:10.3390/cancers15184480_

Round 1

Reviewer 1 Report

* The authors investigated the factors affecting the prognosis of using immune checkpoint inhibitors in patients with non-small cell lung cancer.

* The title must not contain any abbreviation.

* Line 37: Please define NSCLC in its first mention.

* The introduction is well written.

* Line 144: Please, add more details about immunohistochemistry and put the representative figures in the supplementary files.

Reviewer 2 Report

This work is very interesting and meaningful for IO. However,  I have some issues needs to point out.

1. This title is too wide, I suggest the authors only focus on 1-2 part, or change to the clinical, pathological, and gentic related ICI benefit.

2. I suggest the authors add funtional analysis to explain why those signature could predict IO benefit.

3. I also suggest the authors comparison those finding biomarkers with published biomarkers. eg. IMpower150, IMvigor210, IMmotion150 etc.

The authors are native English speakers, so I don't have any other comments about the quality of English language.

Reviewer 3 Report

This is a very good study investigating the predictive potential of gene expression profiling peripheral immune cell counts, and clinical characteristics in patients with NSCLC treated with immunotherapy.

The paper is well written and data are well presented. I particularly liked the choice of DCB as primary endpoint.

Actually, I do not have any specific comment. The main limitation of the study, i.e. the very limited number of patients with available GEP, is acknowledged, but maybe the Authors can do some more effort to discuss it.

English is fine
